# EEG Mental Stress Assessment Using Hybrid Multi-Domain Feature Sets of Functional Connectivity Network and Time-Frequency Features

**DOI:** 10.3390/s21186300

**Published:** 2021-09-20

**Authors:** Ala Hag, Dini Handayani, Thulasyammal Pillai, Teddy Mantoro, Mun Hou Kit, Fares Al-Shargie

**Affiliations:** 1School of Computer Science & Engineering, Taylor’s University, Jalan Taylors, Subang Jaya 47500, Malaysia; alaahmedyahyahag@sd.taylors.edu.my (A.H.); thulasyammal.ramiahpillai@taylors.edu.my (T.P.); 2Faculty of Engineering and Technology, Sampoerna University, Jakarta 12780, Indonesia; teddy.mantoro@sampoernauniversity.ac.id; 3Department of Mechatronic and Biomedical Engineering, Universiti Tunku Abdul Rahman, Bandar Sungai Long, Kajang 43000, Malaysia; munhk@utar.edu.my; 4Department of Electrical Engineering, American University of Sharjah, Sharjah 26666, United Arab Emirates; fyahya@aus.edu

**Keywords:** mental stress, electroencephalography, feature extraction, functional connectivity network, time-frequency features, machine learning

## Abstract

Exposure to mental stress for long period leads to serious accidents and health problems. To avoid negative consequences on health and safety, it is very important to detect mental stress at its early stages, i.e., when it is still limited to acute or episodic stress. In this study, we developed an experimental protocol to induce two different levels of stress by utilizing a mental arithmetic task with time pressure and negative feedback as the stressors. We assessed the levels of stress on 22 healthy subjects using frontal electroencephalogram (EEG) signals, salivary alpha-amylase level (AAL), and multiple machine learning (ML) classifiers. The EEG signals were analyzed using a fusion of functional connectivity networks estimated by the Phase Locking Value (PLV) and temporal and spectral domain features. A total of 210 different features were extracted from all domains. Only the optimum multi-domain features were used for classification. We then quantified stress levels using statistical analysis and seven ML classifiers. Our result showed that the AAL level was significantly increased (*p* < 0.01) under stress condition in all subjects. Likewise, the functional connectivity network demonstrated a significant decrease under stress, *p* < 0.05. Moreover, we achieved the highest stress classification accuracy of 93.2% using the Support Vector Machine (SVM) classifier. Other classifiers produced relatively similar results.

## 1. Introduction

Mental stress has become a catchphrase nowadays, affecting almost everyone, due to the increasing demands in the workplace, life burdens, changing lifestyles, and technological interventions. The long-term effects of stress not only impact health issues, such as heart disease, obesity, diabetes, stroke, and depression [1,2,3], but have economic consequences too. The economic losses can reach up to billions of dollars [4]. Thus, researchers are trying to detect mental stress at its early stage to prevent it from becoming chronic. The evaluation of human psychological stress usually performed using subjective or objective measurement methods. The subjective stress assessment methods used psychological assessment approaches, such as a clinical interview and psychological-based questionnaires, such as the Trier Social Stress Test (TSST) [5,6], Perceived Stress Scale (PSS) [7,8,9], State-Trait Anxiety Inventory (STAI), and Hospital Anxiety and Depression Scale (HADS) [10].

The drawback of subjective methods is that it is subjective to the user’s reported answers, and it only describe the current state of the subject’s stress level. Recent studies have focused on an objective method of physiological assessment, which gives the individuals the freedom to assess their mental stress states without the expert’s intervention and drives more reliable evaluation [11,12,13]. The physiological assessment depends on the body’s reactions towards stress, such as facial expressions [5], blink rate [14], pupil dilation [15], eye gaze, and voice intonation [16]. Several studies have reported that for anyone diagnosed with stress, their body had shown different symptoms by changing the normal activities of these bio-markers: catecholamine, cortisol level, and alpha-amylase enzyme [11,12]. Additionally, during stress, the frontal area of the brain showed high activity of glucose metabolism and blood flow [9,17]. Therefore, several studies utilized different modalities such as functional brain imaging (i.e., functional magnetic resonance imaging (fMRI) and electroencephalography (EEG)) technologies to identify the brain regions and fluctuation of brain activities affected by stress [12,17,18]. The prefrontal cortex (PFC) was the common area that appeared to be sensitive to stress exposure. Moreover, there is an evidence of changes in the autonomic nervous system’s (ANS) activities under stress [19]. Consequently, physiological features of stress from the ANS can be seen as subtle changes in heart rate (HR), heart rate variability (HRV) [6], respiration [20], skin conductance [21], and blood pressure [22]. Currently, the focus is on brain activities as, according to the latest work in neuroscience, it is the main target organ of mental stress due to its responsibility to distinguish between different situations’ contexts (i.e., stressful and threatening or not) [23]. The brain activities are usually analyzed by several tools, such as EEG [24], fMRI [25], positron emission tomography (PET) [26] and other neuroimaging modalities. EEG is a measurement tool that depicts the electrical activities on the brain’s surface. Compared to other modalities, EEG provides high temporal resolution to detect the time variance of changes in the brain’s state [27], is easy to setup, and is commercially available at a lower cost.

To measure stress in real life, researchers used a different approach to stimulate stress in laboratory settings. Several validated stress inducement methods were established such as mental arithmetical tasks [13], negative feedback and time pressure [28], public speaking [29], and noise manifested by music [30]. For the sake of validity and increasing the accuracy of detection, researchers, in many cases, employed a combination of one or more modalities with EEG, such as skin conductance [31], functional near-infrared spectroscopy (fNIRS) [13], and electrocardiography (ECG) [11]. Even though applying additional modulation with EEG improved the accuracy, it is not suitable for home-based applications due to the knowledge base required, lengthy setup time, and the expensive and inconvenient usage for wearable devices. Therefore, current studies suggest that the enhancement of EEG signals could be accomplished by obtaining the optimal features from specific regions of the brain related to the task.

For the aim of finding the relative EEG markers that explain mental stress and increase its detection rate, several studies employed different types of features from the time domain, frequency domain, and time-frequency domain [8,32,33,34,35,36], and several machine learning algorithms have been used to predict the mental stress state, such as SVM [37], K-Nearest Neighbors(KNN) [29,38], LR [1], Feed-Forward Neural Network (FF-NN) [30], Naive Bayes(NB) [9,38], and Random Forest(RF) [39]. In the literature, non-invasive EEG-based stress studies suggested that bio-markers (i.e., alpha, beta, and gamma) in specific brain areas could reveal the mental stress state [18,40,41]. However, no consensus has been reached about the particular established EEG patterns/features that differentiate stress levels, see review [36]. In studies [8,29,42], different frequency band features have been demonstrated to classify stress tasks. The low beta was considered as one approach to recognize mental stress [9]. Similarly, alpha rhythm power at the right PFC was shown to be more discriminative to stress and rest states [28,36]. Another study in [11] showed that the PFC relative gamma power (RG) was more discriminative between stress levels than alpha asymmetry.

Current researchers acknowledge that multi-domain features and multi-channel analyses are required to create an effective information feature space in which a good interpreter can eventually produce effective alarms for the current mental state. As a result, studies presented by Attallah [34] and Hasan [43] revealed that hybrid feature sets from various domains (time domain, frequency domain, and time-frequency domain) may enhance the overall classification of EEG emotion analysis. To the best of our knowledge, no study on fusing such domains with functional connectivity network features has been done. In contrast to the majority of cortical activation features, which focus on a single channel feature, functional connectivity features look for relationships and interactions across different regions of the brain (inter-channel relations). This knowledge aids in a better understanding of how the brain functions and could offer more accurate representations of mental stress states. To address the aforementioned limitation, this study aims to investigate the fusion of functional connectivity network features with cortical activation features from the time and frequency domains to detect mental stress in order to aid in the development of wearable devices. In contrast to prior research, our objective is to combine single-channel features with inter-join channel features (connectivity). Thus, we employ a well-established clinical assessment method using salivary alpha amylase (cortisol measure) to enhance the labeling of the task given. We then propose an objective method based on the machine learning framework to classify stress levels with a minimum number of EEG channels. We present a novel methodology to identify mental stress by investigating the statistical difference between stress and rest conditions. The proposed method is analyzed and evaluated using seven classifiers [36], namely: KNN, RF, Logistic Regression (LR), SVM, classification and regression Decision Tree (CART), Linear Discrimination Analysis (LDA), and NB. The accuracy, precision, recall, and F-score matrices were used to evaluate the performance of the classifiers.

The following section summarizes our contributions in this work.

Developing an experimental protocol to induce two levels of mental stress (stress/rest or control) in a short time, which is important for real-life application.A multi-domain feature set is proposed by fusing features from the time domain, frequency domain, and functional connectivity networks.A feature selection method was implemented to select the most discriminative feature sets.The performance of the proposed method is tested and evaluated using seven machine learning classifiers.

The rest of the paper is organized as follows: Section 2 describes the dataset, protocol setup, and data annotation. Section 3 explains the detailed methodology. In Section 4, the classification evaluation method is presented. In Section 5, results and analysis are discussed in detail. In Section 6, a discussion of the results is provided, and the study’s conclusions are presented in Section 7.

## 2. Dataset and Materials

### 2.1. Participants

In this dataset, the total number of participants was 22 subjects (aged 26 ± 4 with head size of 56 ± 2 cm). All subjects were male right-handed healthy adults having the same culture and background (undergraduate students). The participants were asked about their medical condition to fit the experiment eligibility. Smokers and drug users were excluded due to their effect on the sympathetic nervous system. Moreover, participants must have no history of any physical or mental health problems. Several rules had been imposed on them before starting the experiment. For example, no eating or drinking two hours before the experiment and no physical activity occurred [13]. The experiment time was chosen between 4.00 and 5.30 p.m. to reduce the circadian rhythm’s effects on cortisol collection. The experiment protocol was approved by the institute review board of University Teknologi Petronas.

### 2.2. Stress EEG Measurement and Protocol

The experiment task protocol was based on the Montreal Imaging Stress Task (MIST), which was described in detail in [44]. The task was created using MATLAB and presented using a Graphical User Interface (GUI). It involved a mental arithmetic task (MA) using simple calculation of two-digit integers (ranging from 0 to 100) with operands restricted to +,−, and (/ or *) (example 99/3 − 76 + 51). The answer for each question was displayed in the GUI using a numerical order ranging from ‘0’ to ‘9’, and participants were trained to select the correct answer with a mouse click. The experiment task was performed in three subsequent phases: preparation, rest condition, and stress condition. Each phase is described in detail below. In the preparation phase, participants were given five minutes to practice the MA task, and the average time taken to answer the questions was recorded for each participant, which would later be utilized as a time constraint to induce stress.

In the stress phase, a cap of EEG electrodes was placed on the frontal region of each participant’s scalp, and simultaneous measurement was performed while the participant solved the arithmetic within a time limit (derived based on a 10% reduction from the average time recorded during the preparation phase). Additionally, the average peer performance was displayed on the screen as a real-time performance indicator of subject’s performance compared to other participants. Notification of a negative message for each response that exceeds the time limit or gets the answer wrong, i.e., a message of “Incorrect”, or “Time’s up” being flashed on the screen. The negative feedback was intended to add more stress to the participants.

In the rest phase, the participant was instructed to keep calm and relax while looking at the fixation cross presented at a computer monitor. The presentation of the stress and rest states was in a block design. There was a total of five blocks in each of the stress and rest conditions, as shown in Figure 1. For every block, an arithmetic task popped up for 30 s to induce stress, followed by 20 s of rest. During the 30 s of the stress task, multiple mathematical questions were displayed on the computer monitor based on participants’ response time in answering each question. For the 20 s of a rest condition, the participant looked at the fixation cross in the computer screen as a visual cue for the trial onset.

The EEG signals were recorded using the Discovery 24E system (BrainMaster Technologies Inc, Bedford, OH, USA). The system was equipped with 7-electrodes (Fp1, Fp2, F7, F3, F4, Fz, F) placed on the prefrontal cortex, as shown in Figure 2. The EEG electrodes were referenced to the earlobe electrodes (A1 and A2). The placement of the EEG electrodes was based on the 10–20 system and was sampled at 256 Hz.

### 2.3. Dataset Labelling

The EEG signal has been labeled for each subject based on the cortisol of the salivary amylase level (AAL). During the experiment, two samples of AAL were obtained, as shown in Figure 1. The first AAL sample was collected before starting the experiment task (stress/rest condition) as a baseline. The first AAL result was supposed to show the initial state of the subject as not stressed; otherwise, the subject would be removed from the study. The second AAL sample was collected at the end of the experiment. The data annotation/labeling of the EEG signal was based on the cortisol level; medically, cortisol levels greater than 60 micrograms per decilitre (mcg/dL) indicate that the subject is stressed, while those between 30 and 60 (mcg/dL) are labeled as working brain condition, and those less than 30 (mcg/dL) are labeled as the rest state [44,45].

## 3. EEG Base Mental Stress Analysis Method

This section describes the proposed methodology process for implementing the stress detection method, namely signal preprocessing, feature extraction and selection, and classification, as shown in Figure 3.

### 3.1. Signal Preprocessing

The raw EEG signals were preprocessed using Python and an external MNE package [46]. The raw EEG signals were band-pass filtered using a finite impulse response (FIR) filter with 1 Hz to 35 Hz bandwidth. Since we only measured the frontal cortex, the EEG data were re-referenced to the average reference as suggested by [47]. Consequently, the noise caused by 50/60 Hz of line power was omitted. Furthermore, Fast-ICA has been used to eliminate the associated noise caused by eye blinking called electrooculogram (EOG) artifacts under 4 Hz, muscle artifacts (EMG) with frequency beyond 30 Hz, and heart rate [48]. Fast-ICA has the significant ability of denoising ocular artifacts (OAs) that exist in low frequencies less than 16 Hz, therefore delineating the overlapping frequency bands [49]. The EEG signals were segmented into 1000 ms EEG epochs relative to the target task. The selection of 1000 ms or 256 EEG data points was due to its stationarity at an epoch size of >256 for experiments involved in event-related potential. This number of data points is appropriate to show the stationarity of EEG signals and have been reported in previous EEG studies with a comparable data point [50,51,52]. The baseline was extracted and omitted using the full length of each epoch. Then, all EEG epochs were visually double-checked to eliminate data segments contaminated with noise. Lastly, we identify from the clean EEG signals two mental states (stress and rest). The first 20 s of rest from the first block were considered for the rest state, and another 20 s from the last stress block (block 5) were considered for the stress condition. The two states were labeled based on the results obtained from the cortisol data collection of AAL.

### 3.2. Feature Extraction

Feature extraction is a crucial step in analyzing and classifying EEG signals [43]. Because the EEG signal is a non-stationary and time-varying signal, choosing an appropriate technique to extract useful features that reflect brain activity is critical for reducing dimensional space, improving processing performance, and increasing the detection rate. EEG features can be broadly categorized into single-channel features and multi-channel features. The majority of the existing features are computed on a single channel that involves temporal and or spatial information from a specific brain region, e.g., statistical features, frequency-domain features, e.g., PSD. A few multi-channel features are computed to reflect the relationships between different brain regions, e.g., brain connectivity features. The EEG signal comes from a complex of interconnected brain neurons. Hence, the fusion of the brain connectivity with cortical activation features may provide us with a more exact model of the brain and how its various areas interact with each other. In this paper, both cortical activation features (single-channel features) and functional connectivity network features (multi-channel features) have been employed.

In particular, twelve (single-channel) features were extracted from both the time domain and frequency domain of the cleaned EEG signals for each of the seven channels (Fp1, Fp2, F7, F3, F4, Fz, and F8) located at the prefrontal and frontal region of the brain. Those EEG features were six features per EEG channel from the time domain: kurtosis, peak-to-peak amplitude, skewness, and Hjorth parameters of activity, complexity, and mobility. Likewise, six features per EEG channels were extracted from the frequency domain of relative powers for frequency bands: delta δ, theta θ, alpha α, sigma σ, low beta β, and high beta β. Additionally, a total of 126 features (multi-channel features) were extracted from the connectivity network of all channels. The EEG signal’s length used for feature extraction methods was 40 s (20 s stress, 20 s rest), segmented by the epoch of 1 s, which results in a total of 40 segments, and each segment consists of 210 features per subject for both stress and rest tasks. Table 1 shows the summary of dataset content.

Each of the domain’s features was explained in detail in the following subsections.

#### 3.2.1. Time-Domain Features (TDFs)

The TDFs were calculated from the cleaned EEG signals at each epoch. The TDFs are also called statistical features widely used in the classification of EEG signals to measure the irregularity of signal amplitude in the time domain. Therefore, several studies employed TDFs in emotion [29] and stress classification [37,39]. In this paper, six statistical features were extracted from the time domain, namely: kurtosis, peak amplitude, skewness, and Hjorth parameters of activity, complexity, and mobility. Each of these features was extracted from each channel per subject. The full details of these parameters are given below.

Kurtosis: is the measure of the relative flatness of an EEG signal distribution per segment (epoch), and it is calculated using the equation.
(1)$$Kurtosis=\frac{\frac{1}{T}\sum_{t=1}^{T}{\left(x\left(t\right)-\mu \right)}^{4}}{\sigma^{4}}$$
where *T* is the number of epochs, *x(t)* is time-series sample points, and μ,σ are the mean and standard deviation of the signal.

Skewness: measures the distribution difference between the mean and the median for each variable of epochs.
(2)$$Skewness=\frac{\frac{1}{T}\sum_{t=1}^{T}{\left(x\left(t\right)-\mu \right)}^{3}}{\sigma^{3}}$$

Peak-to-peak amplitude (ptp_amp): the change between the peak of the highest amplitude value and the lowest amplitude value among the various time windows.

Hjorth parameters: three features of Hjorth Parameters (TDHPs), namely activity, mobility, and complexity of the signal, are extracted, which are useful for the quantitative evaluation of an EEG signal and can be expressed as:Hjorth Activity: The activity measure represents the signal power and measures the variance of a time function using the equation.
(3)Activity=var(x(t))
where *x(i)* represents the signal on time.Hjorth Mobility: mobility represents the mean frequency or the proportion of the standard deviation of the signal and is denoted by:
(4)$$Mobility=\sqrt{\frac{var(\frac{dy(t)}{dt})}{Activity(y(t))}}$$
where mobility represents the square root of the variance of the first derivative of the signal *x(t)* divided by the activity.Hjorth complexity: the complexity parameter gives an estimate of the bandwidth of the signal, which indicates the similarity of the shape of the signal to a pure sine wave.
(5)$$Complexity=\sqrt{\frac{Mobility(\frac{dy(t)}{dt})}{Mobility(y(t))}}$$All these extracted features were then fed as an input to the classifiers.

#### 3.2.2. Frequency-Domain Features (FDFs)

In the frequency domain, the multitaper method is used to estimate the power spectral density (PSD) because it provides a more robust spectral estimation than the classical methods and Welch’s periodograms [53]. Compared to Welch’s approach, the multitaper method does not need to identify a window duration because it computes the periodogram on the whole signal and provides a high-frequency resolution and low variance [53].

Multitaper spectrum estimation (MSE): a Nonparametric method used to estimate PSD from a combination of multiple orthogonal tapers (or “windows”). MSE aims to recover the information lost when using a single taper and offers significant performance gains over a nonparametric single taper. The estimator is the average of the K direct spectral estimators, each acting on the whole data record (rather than on a signal segment, as happens in the Welch method) and applying different tapers. Each (partial) estimator is computed by:(6)$${\hat{S}}_{k}={\left|\sum_{i=1}^{N}h_{i,k}X_{i+l-1}e^{-2j\Pi ft\Delta t}\right|}^{2}$$

Let *x(t)*, for *t* = 0, 1,..., *N* 1, be a zero-mean time series with unit sampling and spectral density *S(f)*, Δt is the sampling interval, $h_{i,k}$ is the *k*th data taper, and the bandwidth for Δt is 1 s.

The final estimator is computed as:(7)$${\hat{S}}_{k}=\frac{1}{k}\sum_{k=0}^{k-1}{\hat{S}}_{k}\left(f\right)$$
where *K* is equal to 2NW−1, and 2 W is the normalized bandwidth of the tapers.

The relative power (RP) of six frequency bands: delta (1–4 Hz), theta (4–8 Hz), alpha (8–12 Hz), sigma (12–15 Hz), low beta (15–20 Hz), and high beta (20–30 Hz) were computed from the MSE of PSD. The RP is expressed by divided the specific power band over the total power of all bands and calculated as below:(8)$$RP=\frac{power(selected\_band)}{power(total\_bands)}*100$$

The RP features were then used as an input to the classifiers.

#### 3.2.3. Functional Connectivity Network

The functional connectivity network is generated by measuring the connection between electrode pairs in each frequency band using Phase Locking Value (PLV). The PLV technique, similar to the conventional coherence method, computes the correlation between two pairs of EEG channels in distinct frequency bands. The PLV is an effective measure of brain functional signals due to its ability to quantify locking between the phases of the signals from two distinct electrodes and does not depend on the assumption of stationary signals [54]. Therefore, PLV was proved to be a valid method to investigate task-induced changes in the long-range synchronization of neural activity from EEG data [55]. To calculate the phase-locking value, we extract the instantaneous phase $\phi_{i}^{a}\left(t\right)$ of the analytical signal $x_{i}^{a}\left(t\right)$ of the time series $x_{i}\left(t\right)$.

Then, for each pair (*i*, *j*) of EEG channels, we compute the modulus of the time-averaged phase difference projection onto the unit circle and computed it in Equation (Equation 9):(9)$$VLP_{ij}=\left|\frac{1}{T}\sum_{t}e^{i(\phi_{i}^{a}\left(t\right)-\phi_{j}^{a}\left(t\right))}\right|$$
where *N* is the total number of trials in time series, and $\phi_{i}$ and $\phi_{j}$ are the instantaneous phase values at trial index n. The PLV values range between [0, 1], with 0 indicating no phase synchronization and 1 indicating that there is a fixed relative relationship between the two signals in all trials. Because PLV employs undirected measurement for all electrodes, it is known as symmetric measure (PLV(*k1*, *k2*) = PLV(*k2*, *k1*)).Thus, the direct connection is ignored, and the total number of connections between the EEG channels is measured using:(10)$$N=\frac{k(k-1)}{2}$$
where *k* is the total number of channels. In this paper, the total extracted connectivity features are 126 features (21 features × 6 bands) since we are using only 7 EEG channels. However, we only utilized the PLV with a phase-difference distribution that was significantly different from zero using *t*-test feature selection at *p* < 0.05.

### 3.3. Hybrid Features of Time, Frequency Domain and Connectivity Features

Fusion information from cortical activation (Time and Frequency Domain) and connectivity features might complement each other, giving us a more accurate representation of the brain and how its various regions interact. In this paper, the total features extracted from multi-domain features were 210 features (42 features from time, 42 features from frequency domain, and 126 features from PLV connectivity features), resulting in high-dimensional feature space. Therefore, the significant-features-based channels from the time domain, frequency domain, and connectivity features were identified using a statistical *t*-test with a 95% confident interval and *p* = 0.05 level of significance. Thus, the most significant features from each domain were fused to form a new subset of the significant features from the time domain, frequency domain, and connectivity network. The total significant-features-based channels were 42 (15 from the time domain, 20 from the frequency domain, and 64 from connectivity features ) and used as a new fusion feature set to subsequent classifiers.

## 4. Classification

To classify and evaluate stress levels, three scenarios have been conducted. First, individual feature of the selected channels within each domain was considered as a bio-marker and evaluated separately (i.e., Hjorth complexity, Hjorth mobility, relative alpha, …, etc., see Tables 3–5). In the second scenario, we utilized the features from the selected channels in each domain as a feature vector and classified them separately (i.e., see Figures 8–10). Meanwhile, we fused the features of the selected channels from the three domains: time domain, frequency domain, and connectivity network features into a single feature vector and used them as an input to the ML classifiers (see Figure 11). Several machine learning algorithms have been used for EEG signal analysis to train and predict the features extracted from target EEG tasks. In this paper, seven classifiers, namely LR, RF, LDA, KNN, SVM, DT, and NB, were employed to evaluate the model performance of mental stress recognition based on the scenarios mentioned and provide the researchers with useful information about the effective classifier to be considered in future work. Table 2 shows the classifier’s tuning parameters utilized. More details about the utilized classifiers can be found in our previous study [56]. The extracted features are split into 80% for training and 20% for testing. In each classifier, an independent subject test with 5-fold cross-validation was performed.

The proposed model’s performance has been evaluated using seven classifiers with 5-fold cross-validation and a four-measure matrix. These include accuracy, precision, sensitivity, and F-measure. The equations below show the mathematical formulation for each prediction. The results obtained from the confusion matrix has:True Positives (Tp): The number of labels correctly identified as stress conditions.True Negatives (Tn): The number of labels correctly identified as a rest condition.False Positives (Fp): The number of labels incorrectly identified as stress.False Negatives (Fn): The number of labels incorrectly identified as rest.
(11)$$Accuracy=\frac{Tp+Tn}{Tp+Tn+Fp+Fn}$$
(12)$$Precision=\frac{Tp}{Tp+Fp}$$
(13)$$Sensitivity=\frac{Tp}{Tp+Fn}$$
(14)$$Specificity=\frac{Tn}{Tn+Fp}$$
(15)$$F-measure=2\frac{Precision*Sensitivity}{Precision+Sensitivity}$$

Accuracy denotes the measurement of how many correct predictions were made in the whole dataset in two-class problems, i.e., stress and rest conditions. Precision indicates the correct measure of a positive prediction. Meanwhile, sensitivity refers to the completeness measure of a classifier, measuring the number of true stress conditions that get predicted over whole stress labels in the dataset. Specificity measures the proportion of rest conditions that are correctly identified. The F-measure was used to evaluate the detection result using both sensitivity and precision.

## 5. Result and Analysis

### 5.1. Statistical Analysis

The stress inducement using an arithmetic task under time pressure and negative feedback for 22 subjects was evaluated using a salivary alpha-amylase level and EEG. The stress inducement reported higher levels of salivary alpha-amylase in stress (M = 93.64, SD = 13.99) (KIU/L) compared rest condition (M = 24.45, SD = 4.44) (KIU/L), as shown in Figure 4. The increase in the alpha-amylase level from rest condition to stress condition was significant with a mean *p* < 0.0001. This also correlates with our previous results [13], which revealed a significant difference in alpha-amylase level between the two conditions across all subjects. Therefore, time pressure and negative feedback prove to be reliable for stress induction in the lab.

In EEG signal analysis, an independent-sample *t*-test was conducted to compare stress and rest for each feature-based electrode. The star symbols are used in topographic maps to show the significant electrodes per feature. For time-domain features, Figure 5a shows the mean and standard deviation of the Hjorth complexity, Hjorth mobility, Hjorth activity, kurtosis, peak-to-peak amplitude (ptp_amp), and skewness of EEG signals at 1–30 Hz for stress and rest conditions, which were taken by averaging all subjects’ data for each condition.

The placement of EEG electrodes is coordinated based on the international 10–20 system, as shown in Figure 2. The means of Hjorth complexity, Hjorth activity, and ptp_amp were decreasing from rest condition to stress condition when subjects were exposed to mathematical stressor tasks and increased from rest to stress conditions in Hjorth activity and skewness. This variation in different parameters indicates a further decrease in complexity and ptp_amp from rest to stress conditions but a high increase in the mobility component in the signal. The significant electrodes for time-domain features are shown in Figure 5b, where the color scale represents statistical differences based on *t*-test values.

The total number of features for each channel is six, giving a total of (7 channels × 6 features) 42 features in the time domain. However, only 15 significant features in the time domain that discriminate the rest and stress conditions were selected and used in this study. The topographic T-map for both complexity and mobility features shows the same significant channels of ‘Fp1’, ‘F3’, and ‘F4’; significant Hjorth activity channels were ‘F7’ and ‘F3’. Finally, ptp_amp has four significant channels: ‘F7’, ‘F3’, ‘Fz’, and ‘F8’. Note that skewness and kurtosis were also selected as a useful feature even though there was only one channel, ‘Fp4’ and ‘Fp1’, respectively, for each one, showing a statistically significant difference between the two conditions. This means that the skewness and kurtosis features can offer additional discriminative information between the two conditions. Figure 6 shows the frequency changes on the brain with respect to the stressor test using scale colors of the power distribution of PSD. A statistical analysis of the averaged normalized relative power of the frequency bands (delta,theta,alpha,sigma,low_beta,high_beta) was carried out to demonstrate the difference between rest and stress states. The topographic T-map shows the significant electrodes corresponding to each band with ‘*’ star symbols and the color scale of the T-map. Out of 42 features (7 channels * 6 relative power bands) in the frequency domains, only 20 features were selected as significant features based on *t*-test values for the experiment task of rest and stress conditions.

High beta β(20–30 Hz) showed a significant decrease from the rest condition to the stress condition across all participants with (*p* < 0.001). Consequently, a noticeable significant decrease in the alpha α relative power (8–12 Hz) was found in the right cortex of the frontal area for the mathematical stressor tests from the rest condition to the stress condition. Likewise, theta θ (4–8 Hz) relative power indicated a slightly significant increase in stress conditions compared to rest conditions. The overall statistical analysis of the average relative power was shown to be discriminative among stress and rest conditions in all significant electrodes with *p* < 0.0001 in alpha and high beta and *p* < 0.001 in delta and sigma (12–15 Hz) with *p*-value 0.05. Interestingly, the prefrontal right cortex channels (‘Fp1’, ‘Fp2’, and ‘F4’) were shown to be more significant in most relative power bands to distinguish rest and stress conditions.

Similarly, Figure 7 shows the functional connectivity network features of PLV, which measures the changes (increase/decrease) in the connectivity network between two pairs of channels. The PLV was extracted from six frequency bands of both tasks (rest/stress). The significant channels denoted either an increase or decrease (*+/*−) in the connectivity network measurements from the rest condition to the stress condition. From Figure 7, it can be seen that significant functional connectivity networks in delta and alpha decreased from the rest condition to the stress condition. On the other hand, high beta shows an increase in the connectivity network from the rest condition to the stress conditions. Other bands showed increases and decreases in the connectivity network between different brain regions. The significant discrimination connectivity features between stress and rest conditions were selected using a *t*-test and fused with other significant features from the time and frequency domains.

### 5.2. Classification Results

The overall classification performance results in terms of the average accuracy/ precision/recall/f-score and standard deviations of the proposed methods with the types of classifiers are presented in Table 3 for time-domain features, in Table 4 for frequency-domain features, and Table 5 for connectivity network features. Those average accuracies were evaluated for the seven classification algorithms with respect to the number of channels selected by the *t*-test. From the time-domain features in Table 3, we obtain the following significant findings:•The best classification accuracy was obtained by using the peak-to-peak amplitude feature with four channels (‘F7’, ‘F3’, ‘Fz’, and ‘F8’) of the frontal region with a mean accuracy of 79.4% using Random Forest and 76.1% using SVM. Meanwhile, the rest of the classifiers achieved an average accuracy of 75% for ptp_amp.•The Hjorth parameters of complexity, mobility, and activity achieved a result of an average of 69.1%, 71.5%, and 71.8% using KNN, SVM, and NB, respectively. The significant channels of Hjorth complexity and mobility were located in the prefrontal cortex (‘Fp1’, ‘F3’, and ‘F4’), while Hjorth activity had only two significant channels that were selected from the left frontal cortex of (‘F7’ and ‘F3’).•Features with a low number of channels tends to achieve low accuracy due to low spatial resolution. Kurtosis and skewness got one significant channel for each and obtained a maximum average accuracy of 55% and 56%, respectively, for ‘Fp1’ and ‘F4’.

Furthermore, significant findings from frequency-domain features in Table 4 were elaborated below.

•The highest average accuracy achieved by the relative power of the high beta band (20–30 Hz) with significant channels ‘Fp1’, ‘Fp2’, ‘F3’, and ‘F4’ was 73% accuracy with KNN and 71% with both RF and SVM.•For the lower frequencies of delta (1–4 Hz) and theta (4–8 Hz), the significant selected channels were located in the prefrontal and middle frontal cortex area of the scalp—‘Fp1’, ‘Fp2’, ‘F3’, and ‘F4’. Both achieved an average accuracy of 68% using SVM and KNN, respectively.•The lowest accuracy obtained in frequency-band features was 52.4% from sigma relative power with only one channel of ‘F7’.•Likewise, the average accuracy of alpha relative power and low betas were 63.4% and 64.5% with KNN and LDA, respectively.

The overall observation of significant selected channels for both time-domain and frequency-domain features was observed in ‘Fp1’, ‘Fp2’, ‘F3’, and ‘F4’.

Table 5 presents the classification performance of each significant PLV of the connectivity frequency bands. The highest accuracy achieved by PLV bands were 0.752 ± 0.144, 0.734 ± 0.145, and 0.719 ± 0.177 for PLV’s of delta, high beta, and alpha, respectively, using LDA. The rest of the PLV bands got an average accuracy of 0.65 ± 0.12.

We further classified each subset of significant features of the time domain, frequency domain, and connectivity features as feature vectors and passed them to the classifiers. Figure 8 shows the average accuracy of 15 significant time-domain features and achieved a high accuracy of 81.4% and 80% using RF and SVM, respectively, while other classifiers achieved an average of 76.4%. Figure 9 represents the average accuracies of 20 significant features of relative powers in the frequency domain. The highest accuracy of 80% was obtained by SVM, and 74% was the average accuracy of the other classifiers. Similarly, Figure 10 shows the results of 29 significant features from the connectivity network of PLV, and the average performance accuracy obtained was 88% with SVM and RF, while the rest of the classifiers achieved an average of 84%. Meanwhile, Figure 11 presents the average classification result of 64 hybrid significant features from the time domain, frequency domain, and functional connectivity network, as shown in Table 3, Table 4 and Table 5.

Figure 12 demonstrates the comparison of classification accuracy of each feature’s subset domain (time-domain feature, frequency domain, connectivity network feature) as well as after their fusion. In summary, these results show that SVM achieved the best classification performance when fusing connectivity features with cortical connectivity features, scoring 93.2%, 92.4%, 92.5%, and 92.1% for accuracy, precision, recall, and f1-score, respectively. Overall, fusing the multi-domain feature set from cortical and connectivity features improves the classification performance by 13% compared to a single subset domain alone.

## 6. Discussion

This study has presented a methodological approach based on the fusion of multi-domain EEG features and ML for the sake of mental stress classification. To the best of our knowledge, this is the first EEG study on stress that fused functional connectivity features with temporal and spectral features. For this aim, an experimental paradigm based on MIST was designed to induce mental stress and rest conditions using mathematical task with time pressure and negative feedback. For the comparison between the two conditions, a valid objective measurement using the alpha-amylase level (AAL) was collected from the saliva of each subject under both conditions (rest/stress) and quantitatively analyzed, as shown in Figure 4. We found that induced stress revealed a significant difference in AAL between the rest and stress conditions across all subjects, with a considerable increase in AAL from the rest condition to the stress condition. This study correlates to prior findings [13,57] of utilizing arithmetical tasks to induce mental stress in the laboratory setting.

Compared to previous stress detection methods, the main contributions of the proposed method are exploiting the different feature extraction methods and analyzing the significant corresponding channels. For the identification of mental stress in EEG, three scenarios for feature analysis were conducted:

The first scenario analyzed features of the time domain, frequency domain, and functional connectivity network separately, as shown in Table 3 and Table 4 and Figure 7. Prior to the analysis, only significant channels were selected for classification. The selection of significant channels in all types of features was based on a statistical *t*-test at *p* < 0.05. The second scenario was based on combining the significant features within the time domain, frequency domain, and connectivity features of PLV to form subset feature vectors for classification (Figure 8 and Figure 9). The third scenario was based on the fusion of the significant features from all domains (time domain, frequency domain, and connectivity network features) to form a single hybrid subset feature vector for subsequent classifiers.

In particular, for the temporal features of Hjorth complexity, Hjorth activity, ptp_amp, and kurtosis, we found a significant decrease from the rest condition to the stress condition. The decrease in the temporal activities within stress conditions indicates that participants experience difficulties in engaging with the MA task. In fact, the greater the complexity value is, the more active the brain is. Previous studies have found that higher complexity meant increased behavioral performance [58,59]. In line with that, the decreased complexity in our study is a sign of decreased behavioral performance (accuracy of detection) due to stress. It should be noted, however, that the decrease in brain activity/complexity in our study was localized to a certain brain region. For example, when the temporal features of Hjorth complexity are considered, the left frontal region at ‘Fp1’ and ‘F3’ is highly reduced under stress. This reduction is consistent with the previous emotion study that utilized videos to induce negative emotions in the participants [60]. Likewise, when the complexity and skewness are considered, the right frontal region at ‘FP2’ and ‘F4’ is highly reduced. This is also consistent with our previous studies that utilized simple arithmetic tasks with time pressure to induce stress [13,27].

On the other hand, we found that the relative EEG power in theta, alpha, sigma, and beta showed a significant increase from the rest condition to the stress condition at a particular region of the brain. Considering all of the relative powers together, we found that the right hemisphere was highly sensitive to stress exposure. This confirms that negative emotions are induced under stress. This is in line with previous studies that showed when the stress level increased, the alpha power increased across the frontal cortex [44,57,61,62]. Likewise, the increase in relative beta power in our study is also consistent with previous studies that utilized driving and public speaking as stressors in their studies [39,63]. These findings demonstrate the potential of using temporal, spectral, and connectivity features in finding patterns associated with mental stress, as demonstrated in our previous studies on stress and control states [64,65].

We further analyze the classification accuracy of stress based on the first scenario using CAR, KNN, LDA, LR, NB, RF, and SVM classifiers. The temporal features of the peak-to-peak amplitude of the significant channels—‘F7’, ‘F3’, ‘Fz’, and ‘F8’—showed the highest classification accuracy of 79.4% using SVM. Meanwhile, the frequency-domain features of the high beta band (20–30 Hz) at significant channels of ‘Fp1’, ‘Fp2’, ‘F3’, and ‘F4’ achieved the highest classification accuracy of 73% using KNN.

Meanwhile, in the second scenario (domain feature subset analysis), we observed a 2%, 7%, and 13% improvement in classification accuracy in the time domain, frequency domain, and connectivity features, respectively, when compared to the first scenario. Particularly, the high accuracies of 81.4%, 80% and 88% were achieved when using the significant feature subset of the 15 time-domain features, 20 frequency-domain features, and 29 connectivity network features, respectively. It is noteworthy that the subset of connectivity features outperformed other domains in classifying mental stress. Our findings are consistent with prior functional connectivity research, which has shown that functional connection is more reflective of the mental task performance [66].

Additionally, in the third scenario, fusing significant features of the time domain, frequency domain, and connectivity features of PLV (a total of 64 features) improved the overall accuracy of detecting the rest/stress condition with the highest accuracy of 93.2% obtained using SVM. As expected, the improvements in the classification performance support the hypothesis that fusing multi-domain features may provide complementary information for better stress detection. In general, the proposed method of selecting a significant EEG-channel-based feature yield a total reduction of feature space of almost 60%, with 64 significant features out of 210 features.

The overall classifiers’ performance depends significantly on relevant EEG features and the selected channel related to the given task. Previous studies have found that using a large number of EEG channels could provide high resolution and improve accuracy; however, they have inherited issues, such as cost and applicability, particularly outside laboratories. Subhani [57] discusses the identification of stress using 19 EEG channels with features of absolute power, relative power (RP), coherence, phase lag, and amplitude asymmetry, and they reported high accuracy of 94.58%; yet, high dimensionality existed when the 190-feature vector was used. This high accuracy could be interpreted as the result of using a high spatial resolution of 19 EEG channels. However, in this study, the maximum accuracy was 93.2% with the 64-feature vector. Although the accuracy was slightly lower than the one reported by [57,63], this study efficiently reduced feature space with high performance.

This study confirmed that the fusion of temporal, spectral, and connectivity features significantly improved mental stress classification accuracy. Although the study was informative for the mental stress classification, it had a few limitations. First, we only reported the results of EEG feature extraction at an epoch length of one second, which corresponds to 256 EEG data points. Future studies may consider reporting the results of epochs with more than one second, i.e., in the range of one to ten seconds. Second, this study was constructed on fusion functional connections using PLV with cortical features. Other connectivity features, such as the Phase Lock Index (PLI), Partial Directed Coherence (PDC), and Directed Transfer Function (DTF) [54], were not included in the study. Third, while we conducted statistical analysis with a *t*-test to select the brain regions relevant to mental stress in this study, different methods for detecting mental stress using feature-based channel selection (e.g., swarm intelligence) should be considered in future work to reduce the high dimensionality and select the optimal feature set. Finally, throughout the experiment, we found that the SVM outperformed other classifiers in terms of classification performance using the selected hyperparameters, but using algorithm optimization for finding optimal parameters could improve the overall performance.

In essence, the proposed framework empirically proved the possibility of having significant channels corresponding to each feature while eliminating the redundancy and ignoring un-relevant channels. Then, it could be suggested that EEG signals have the potential to be reliable for identifying stress for home-based applications with an optimal number of channels and the relevant features. However, multiple methods for detecting mental stress using feature-based channel selection should be considered in future work.

## 7. Conclusions

This paper aims to find feature sets that would distinguish the stress and non-stress conditions using seven frontal EEG channels. EEG’s features from the time domain, frequency domain, functional connectivity network, and all three fused together were investigated. Seven classifiers were used to evaluate the performance of each feature set before and after fusion. The highest accuracy of 93.2% was achieved using hybrid features with the SVM classifier. In comparison, the evaluation performance of the time domain, frequency domain, and connectivity feature subsets were 81.4%, 80%, and 88% respectively. The results demonstrated that the proposed method of fusing the connectivity network with temporal and spectral features was capable of detecting mental stress state with high classification performance. The overall results support developing a real-time system for stress measurement and analysis.

## Figures and Tables

**Figure 1 sensors-21-06300-f001:**
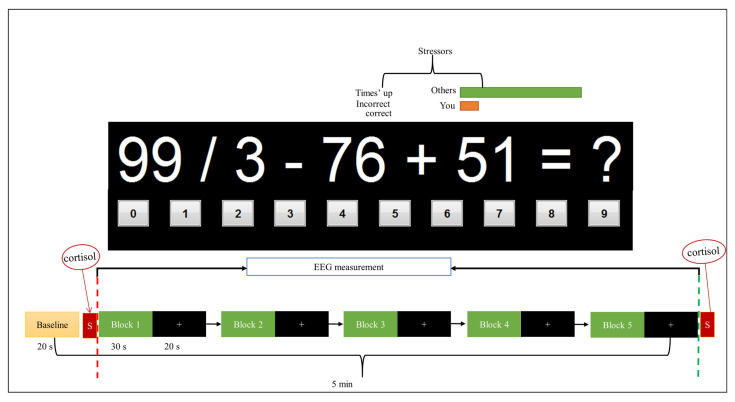
Experiment block design. A total of five active blocks for each task with salivary alpha amylase (SAA) cortisol was collected before and after the stress task and presented by the letter S with a red background. For each block, arithmetic tasks are given for the 30 s followed by 20 s of rest. The red dashed line marks the start of the task, and the green dashed line marks the end of the task (the marking is done at every block).

**Figure 2 sensors-21-06300-f002:**
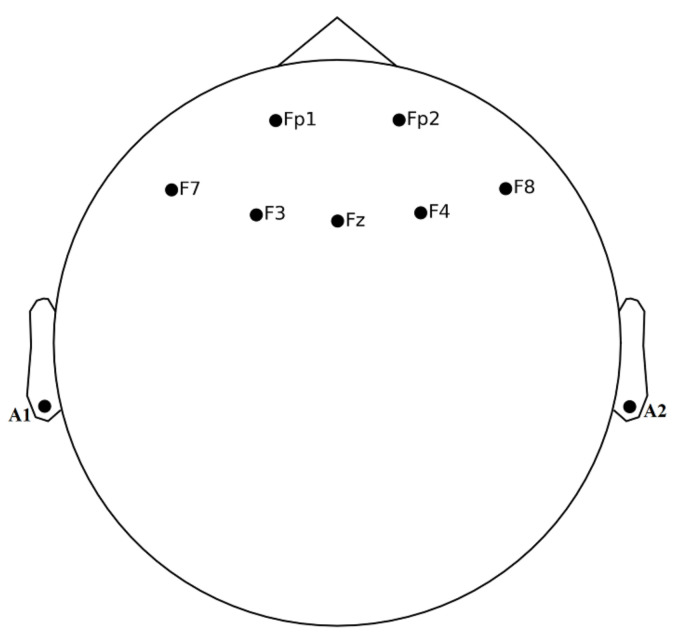
EEG Channels’ Position on Scalp.

**Figure 3 sensors-21-06300-f003:**
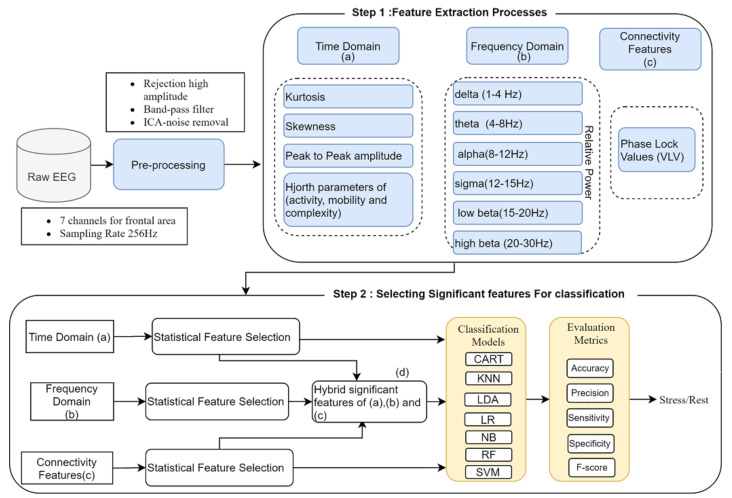
Proposed ML methodology flow chart for mental stress state recognition.

**Figure 4 sensors-21-06300-f004:**
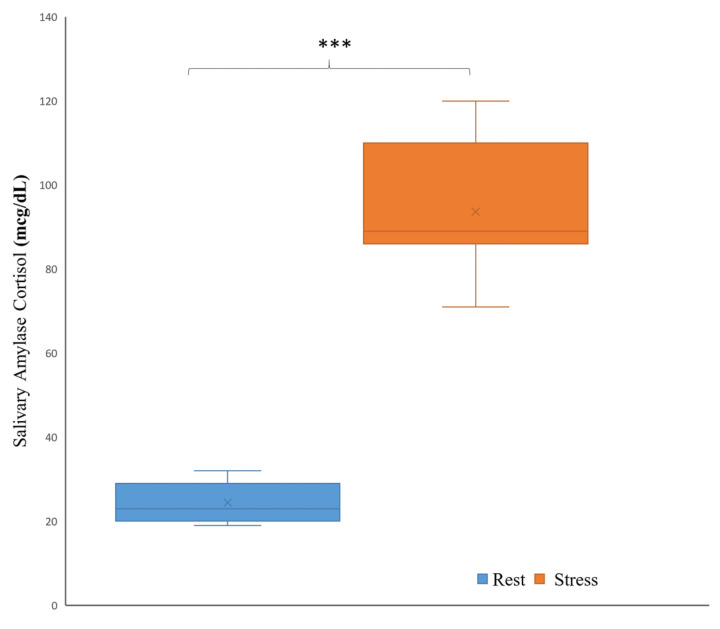
The mean and standard deviation of the salivary amylase cortisol measured by (mcg/dL) for rest and stress conditions.

**Figure 5 sensors-21-06300-f005:**
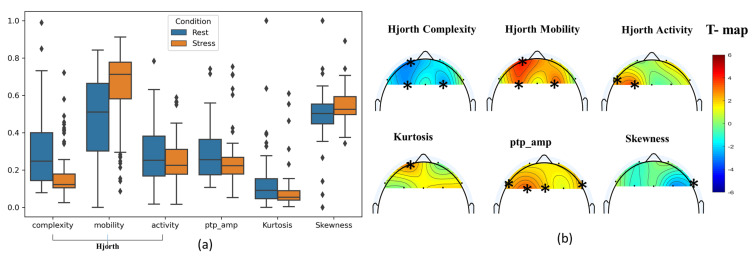
(**a**) The mean and standard deviation using scatter for time-domain features of the Hjorth complexity, Hjorth mobility, Hjorth activity, kurtosis, peak-to-peak amplitude (ptp_amp), and skewness of EEG signals at 1–30 Hz for stress and rest conditions. The difference between stress and rest is shown using T-maps in (**b**). The star (*) symbols denote statistically significant electrodes using topographic maps (two-sample *t*-test; *p* < 0.01, Bonferroni correction).

**Figure 6 sensors-21-06300-f006:**
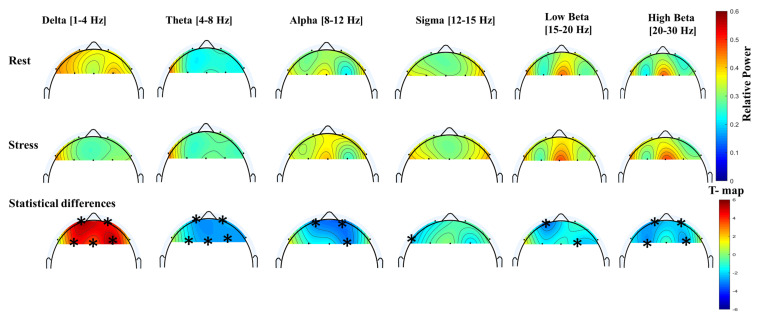
The mean topographic maps for relative bands power of delta, theta, alpha, sigma, low beta, and high beta at 1–30 Hz for rest and stress conditions. The difference between stress and rest relative powers are shown using T-maps. The star (*) symbols denote to the significant electrodes related to specific feature (two-sample *t*-test; *p* < 0.01, Bonferroni correction).

**Figure 7 sensors-21-06300-f007:**
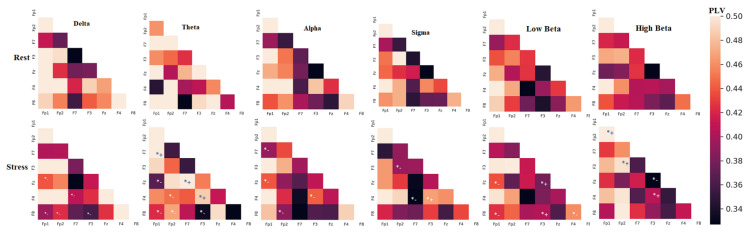
The PLV connectivity network among EEG channel pairs over all trails for rest and stress condition. The star (*) symbol denotes the significant connections between electrodes selected by the *t*-test.

**Figure 8 sensors-21-06300-f008:**
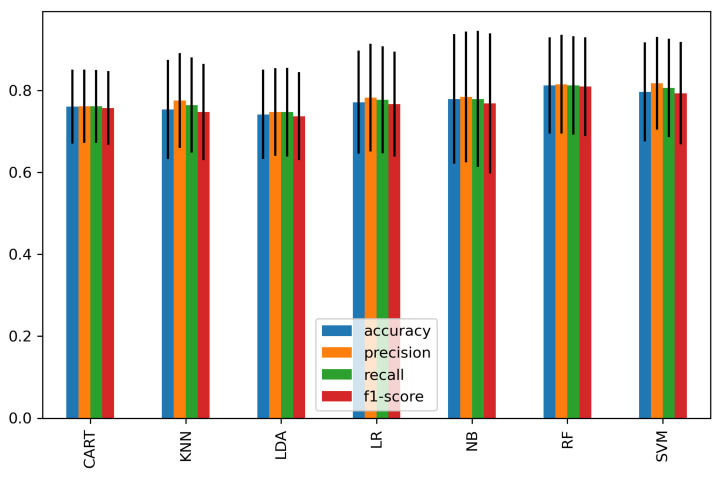
The average classification performance and standard deviation σ of 15 significant features of the time domain. The vertical line indicates σ.

**Figure 9 sensors-21-06300-f009:**
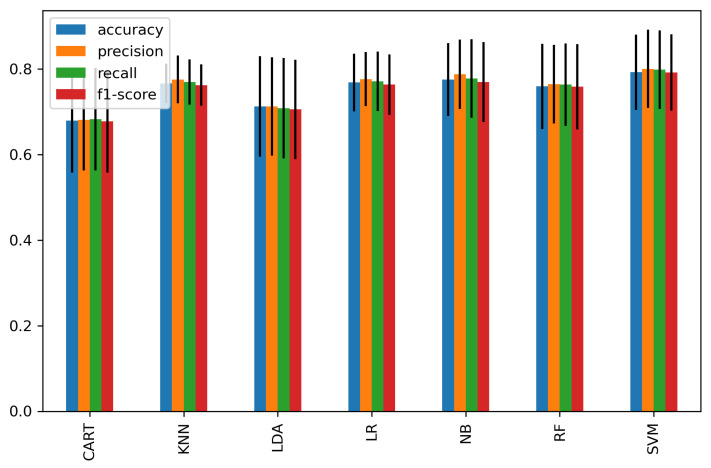
The average classification performance and standard deviation of 20 significant features from the frequency domain.

**Figure 10 sensors-21-06300-f010:**
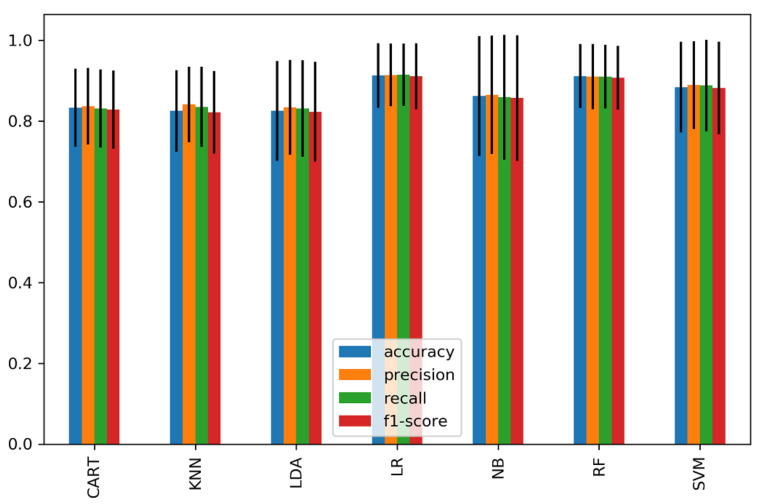
The average classification performance and standard deviation of all significant connectivity network features of PLV.

**Figure 11 sensors-21-06300-f011:**
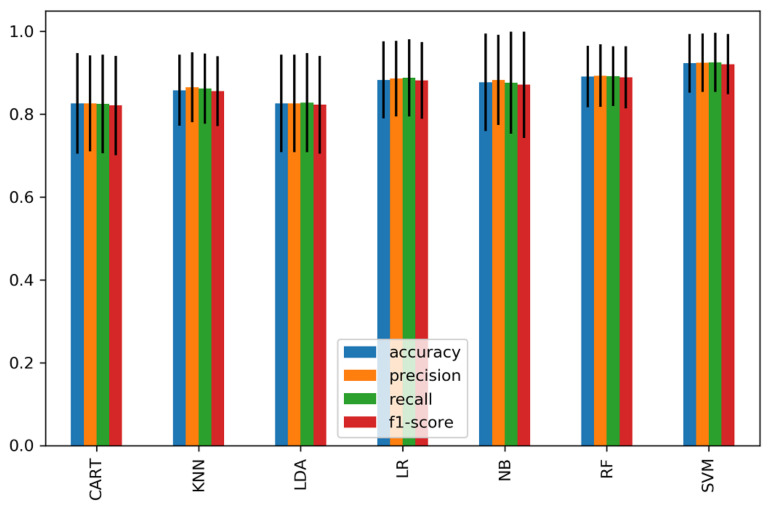
The average classification performance and standard deviation of hybrid features consisting of 42 significant features from the time domain, frequency domain, and connectivity network.

**Figure 12 sensors-21-06300-f012:**
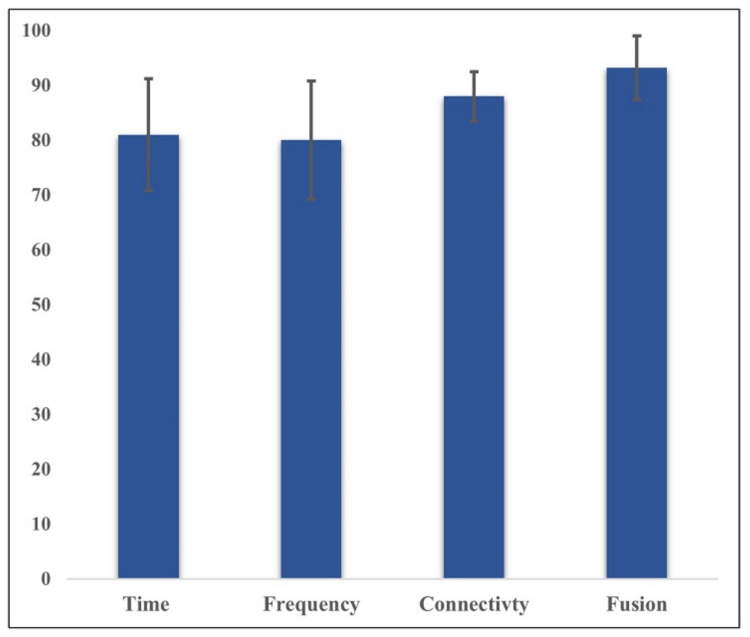
A summary comparison between the average accuracy of single subset domain features (time domain, frequency domain, connectivity network features) and the fusion of all three using SVM.

**Table 1 sensors-21-06300-t001:** A summary of Dataset Structure Content.

Name	Array Shape	Array Content
Data	40 × 7 × 256	Trails × channels × samples (256 Hz × 1 s)
Label	40 × 2	Trail × label (stress, rest)

**Table 2 sensors-21-06300-t002:** Default Parameters for Classification Techniques.

No.	Classifier	Default Value
1	SVM	C = 1.0, Kernal = Radial Basis Function (RBF), 1.0 × 10^−3^
2	KNN	K = 5, distance function = euclidean distance
3	NB	variance_smoothing = 1 × 10^−9^
4	RF	n_estimators = 100 trees, criterion = ‘gini’
5	DT	criterion = ‘gini’
6	LR	penalty = ‘l2’, *, tolerance = 0.0001, C = 1.0
7	LDA	solver = ‘Singular value decomposition (svd)’, tolerance = 0.0001

**Table 3 sensors-21-06300-t003:** The average accuracies and standard deviations of significant time-domain features from the selected channels with the seven classifiers.

Features	Sig. Channels (*p* < 0.001)	Performance	CART	KNN	LDA	LR	NB	RF	SVM
Hjorth complexity	‘Fp1’, ‘F3’, ‘F4’	accuracy	0.644 ± 0.146	0.691 ± 0.119	0.671 ± 0.103	0.659 ± 0.118	0.682 ± 0.114	0.671 ± 0.145	0.677 ± 0.124
		precision	0.653 ± 0.151	0.715 ± 0.118	0.680 ± 0.109	0.672 ± 0.126	0.696 ± 0.116	0.673 ± 0.15	0.694 ± 0.128
		recall	0.650 ± 0.149	0.701 ± 0.117	0.675 ± 0.107	0.668 ± 0.123	0.689 ± 0.117	0.672 ± 0.146	0.688 ± 0.127
		f1-score	0.641 ± 0.149	0.688 ± 0.119	0.667 ± 0.108	0.656 ± 0.119	0.678 ± 0.118	0.667 ± 0.148	0.675 ± 0.125
Hjorth mobility	‘Fp1’, ‘F3’, ‘F4’	accuracy	0.706 ± 0.127	0.702 ± 0.158	0.705 ± 0.122	0.516 ± 0.072	0.709 ± 0.129	0.712 ± 0.146	0.715 ± 0.140
		precision	0.711 ± 0.128	0.721 ± 0.159	0.713 ± 0.114	0.519 ± 0.081	0.715 ± 0.126	0.718 ± 0.140	0.729 ± 0.135
		recall	0.704 ± 0.124	0.711 ± 0.155	0.711 ± 0.117	0.519 ± 0.078	0.711 ± 0.124	0.715 ± 0.136	0.725 ± 0.137
		f1-score	0.703 ± 0.128	0.698 ± 0.160	0.702 ± 0.123	0.609 ± 0.039	0.707 ± 0.128	0.708 ± 0.147	0.710 ± 0.144
Hjorth activity	‘F7’, ‘F3’	accuracy	0.541 ± 0.000	0.706 ± 0.184	0.655 ± 0.173	0.511 ± 0.001	0.718 ± 0.168	0.540 ± 0.120	0.688 ± 0.117
		precision	0.301 ± 0.000	0.714 ± 0.189	0.659 ± 0.179	0.431 ± 0.000	0.727 ± 0.164	0.387 ± 0.032	0.699 ± 0.183
		recall	0.371 ± 0.000	0.705 ± 0.183	0.656 ± 0.177	0.372 ± 0.000	0.723 ± 0.166	0.394 ± 0.013	0.691 ± 0.177
		f1-score	0.321 ± 0.000	0.699 ± 0.190	0.648 ± 0.180	0.321 ± 0.000	0.708 ± 0.177	0.386 ± 0.024	0.681 ± 0.176
Kurtosis	‘Fp1’	accuracy	0.549 ± 0.118	0.531 ± 0.089	0.515 ± 0.130	0.517 ± 0.129	0.516 ± 0.136	0.549 ± 0.118	0.518 ± 0.136
		precision	0.551 ± 0.117	0.539 ± 0.107	0.539 ± 0.163	0.541 ± 0.163	0.511 ± 0.200	0.551 ± 0.117	0.524 ± 0.184
		recall	0.549 ± 0.115	0.536 ± 0.100	0.526 ± 0.139	0.526 ± 0.139	0.523 ± 0.147	0.549 ± 0.115	0.521 ± 0.153
		f1-score	0.544 ± 0.119	0.524 ± 0.095	0.504 ± 0.131	0.509 ± 0.129	0.484 ± 0.151	0.544 ± 0.119	0.498 ± 0.153
PTP_AMP	‘F7’, ‘F3’, ‘Fz’, ‘F8’	accuracy	0.758 ± 0.127	0.735 ± 0.146	0.754 ± 0.126	0.401 ± 0.000	0.745 ± 0.152	0.794 ± 0.122	0.761 ± 0.145
		precision	0.764 ± 0.125	0.745 ± 0.144	0.767 ± 0.120	0.301 ± 0.000	0.752 ± 0.154	0.798 ± 0.119	0.766 ± 0.146
		recall	0.759 ± 0.125	0.738 ± 0.141	0.761 ± 0.121	0.371 ± 0.000	0.745 ± 0.162	0.796 ± 0.121	0.766 ± 0.146
		f1-score	0.754 ± 0.126	0.732 ± 0.146	0.752 ± 0.127	0.321 ± 0.000	0.734 ± 0.162	0.790 ± 0.123	0.756 ± 0.146
Skewness	‘F4’	accuracy	0.561 ± 0.108	0.538 ± 0.101	0.490 ± 0.141	0.467 ± 0.119	0.490 ± 0.120	0.561 ± 0.108	0.483 ± 0.125
		precision	0.566 ± 0.112	0.546 ± 0.104	0.493 ± 0.148	0.467 ± 0.132	0.503 ± 0.153	0.566 ± 0.112	0.492 ± 0.154
		recall	0.564 ± 0.110	0.546 ± 0.101	0.491 ± 0.145	0.467 ± 0.125	0.491 ± 0.128	0.564 ± 0.110	0.493 ± 0.134
		f1-score	0.559 ± 0.107	0.533 ± 0.103	0.488 ± 0.140	0.461 ± 0.124	0.480 ± 0.118	0.559 ± 0.107	0.471 ± 0.134

**Table 4 sensors-21-06300-t004:** The summary of average accuracies and standard deviations of significant frequency-domain features from the selected channels with the seven classifiers.

Band	Sig. Channels (*p* < 0.001)	Performance	CART	KNN	LDA	LR	NB	RF	SVM
Delta	‘Fp1’, ‘Fp2’, ‘F3’, ‘Fz’, ‘F4’	accuracy	0.631 ± 0.127	0.659 ± 0.152	0.680 ± 0.123	0.488 ± 0.069	0.672 ± 0.128	0.675 ± 0.113	0.682 ± 0.144
		precision	0.639 ± 0.122	0.679 ± 0.150	0.692 ± 0.120	0.488 ± 0.076	0.682 ± 0.128	0.686 ± 0.102	0.694 ± 0.145
		recall	0.635 ± 0.121	0.674 ± 0.146	0.689 ± 0.121	0.495 ± 0.066	0.679 ± 0.129	0.682 ± 0.104	0.690 ± 0.145
		f1-score	0.627 ± 0.127	0.652 ± 0.152	0.679 ± 0.123	0.481 ± 0.079	0.672 ± 0.128	0.673 ± 0.113	0.677 ± 0.145
Theta	‘Fp1’, ‘Fp2’, ‘F3’, ‘Fz’, ‘F4’	accuracy	0.614 ± 0.088	0.619 ± 0.087	0.679 ± 0.142	0.473 ± 0.059	0.655 ± 0.118	0.643 ± 0.107	0.656 ± 0.135
		precision	0.614 ± 0.091	0.623 ± 0.100	0.683 ± 0.146	0.471 ± 0.064	0.662 ± 0.132	0.646 ± 0.107	0.659 ± 0.140
		recall	0.614 ± 0.091	0.616 ± 0.095	0.671 ± 0.138	0.475 ± 0.060	0.652 ± 0.119	0.645 ± 0.104	0.655 ± 0.140
		f1-score	0.611 ± 0.091	0.607 ± 0.097	0.669 ± 0.138	0.464 ± 0.068	0.647 ± 0.122	0.640 ± 0.103	0.648 ± 0.136
Alpha	‘Fp1’, ‘Fp2’, ‘F4’	accuracy	0.622 ± 0.107	0.634 ± 0.149	0.634 ± 0.089	0.481 ± 0.070	0.619 ± 0.123	0.606 ± 0.096	0.622 ± 0.137
		precision	0.627 ± 0.108	0.646 ± 0.153	0.640 ± 0.096	0.477 ± 0.076	0.622 ± 0.126	0.608 ± 0.094	0.630 ± 0.141
		recall	0.622 ± 0.101	0.639 ± 0.144	0.633 ± 0.091	0.477 ± 0.076	0.615 ± 0.124	0.607 ± 0.093	0.626 ± 0.136
		f1-score	0.618 ± 0.105	0.628 ± 0.151	0.630 ± 0.091	0.474 ± 0.073	0.611 ± 0.122	0.602 ± 0.095	0.618 ± 0.136
Sigma	‘F7’	accuracy	0.524 ± 0.101	0.485 ± 0.153	0.467 ± 0.143	0.422 ± 0.083	0.520 ± 0.145	0.524 ± 0.101	0.494 ± 0.166
		precision	0.532 ± 0.121	0.491 ± 0.161	0.470 ± 0.147	0.422 ± 0.091	0.522 ± 0.153	0.532 ± 0.120	0.501 ± 0.181
		recall	0.528 ± 0.111	0.490 ± 0.156	0.473 ± 0.145	0.422 ± 0.087	0.519 ± 0.150	0.528 ± 0.111	0.499 ± 0.175
		f1-score	0.521 ± 0.103	0.475 ± 0.156	0.463 ± 0.144	0.415 ± 0.085	0.510 ± 0.151	0.521 ± 0.103	0.480 ± 0.169
Low Beta	‘Fp1’, ‘F4’	accuracy	0.608 ± 0.081	0.597 ± 0.134	0.646 ± 0.087	0.494 ± 0.050	0.626 ± 0.123	0.574 ± 0.100	0.617 ± 0.104
		precision	0.612 ± 0.082	0.601 ± 0.136	0.649 ± 0.091	0.489 ± 0.060	0.635 ± 0.124	0.580 ± 0.101	0.624 ± 0.106
		recall	0.610 ± 0.082	0.598 ± 0.136	0.647 ± 0.090	0.492 ± 0.055	0.630 ± 0.120	0.578 ± 0.101	0.618 ± 0.101
		f1-score	0.605 ± 0.081	0.592 ± 0.134	0.643 ± 0.091	0.488 ± 0.060	0.626 ± 0.123	0.571 ± 0.100	0.613 ± 0.103
High Beta	‘Fp1’, ‘Fp2’, ‘F3’, ‘F4’	accuracy	0.658 ± 0.120	0.729 ± 0.122	0.726 ± 0.133	0.505 ± 0.069	0.734 ± 0.136	0.713 ± 0.118	0.714 ± 0.115
		precision	0.660 ± 0.105	0.736 ± 0.125	0.733 ± 0.137	0.505 ± 0.071	0.735 ± 0.135	0.716 ± 0.115	0.718 ± 0.109
		recall	0.656 ± 0.104	0.731 ± 0.119	0.726 ± 0.129	0.509 ± 0.065	0.732 ± 0.134	0.713 ± 0.112	0.716 ± 0.111
		f1-score	0.651 ± 0.111	0.727 ± 0.124	0.723 ± 0.132	0.499 ± 0.078	0.731 ± 0.137	0.707 ± 0.119	0.711 ± 0.116

**Table 5 sensors-21-06300-t005:** The summary of average accuracies and standard deviations of PLV’s significant connectivity network features from each band.

Features	Performance	CART	KNN	LDA	LR	NB	RF	SVM
Delta_PLV	accuracy	0.678 ± 0.107	0.684 ± 0.141	0.752 ± 0.144	0.657 ± 0.161	0.718 ± 0.166	0.717 ± 0.150	0.724 ± 0.179
	precision	0.678 ± 0.111	0.691 ± 0.153	0.757 ± 0.142	0.661 ± 0.161	0.720 ± 0.162	0.719 ± 0.152	0.731 ± 0.182
	recall	0.677 ± 0.109	0.681 ± 0.143	0.753 ± 0.140	0.658 ± 0.160	0.717 ± 0.162	0.713 ± 0.146	0.725 ± 0.177
	f1-score	0.674 ± 0.108	0.678 ± 0.143	0.749 ± 0.145	0.653 ± 0.161	0.712 ± 0.165	0.711 ± 0.148	0.717 ± 0.181
Theta_PLV	accuracy	0.606 ± 0.172	0.681 ± 0.137	0.683 ± 0.129	0.631 ± 0.163	0.683 ± 0.145	0.629 ± 0.170	0.651 ± 0.179
	precision	0.614 ± 0.172	0.681 ± 0.139	0.687 ± 0.130	0.631 ± 0.168	0.690 ± 0.144	0.636 ± 0.174	0.651 ± 0.180
	recall	0.612 ± 0.172	0.674 ± 0.136	0.685 ± 0.128	0.627 ± 0.164	0.681 ± 0.146	0.631 ± 0.169	0.649 ± 0.178
	f1-score	0.604 ± 0.174	0.673 ± 0.137	0.681 ± 0.129	0.624 ± 0.164	0.676 ± 0.145	0.627 ± 0.169	0.646 ± 0.178
Alpha_PLV	accuracy	0.686 ± 0.140	0.662 ± 0.148	0.719 ± 0.177	0.657 ± 0.180	0.710 ± 0.162	0.699 ± 0.142	0.691 ± 0.143
	precision	0.686 ± 0.139	0.664 ± 0.154	0.719 ± 0.182	0.659 ± 0.181	0.709 ± 0.161	0.701 ± 0.142	0.696 ± 0.146
	recall	0.686 ± 0.139	0.657 ± 0.149	0.713 ± 0.181	0.658 ± 0.181	0.706 ± 0.160	0.703 ± 0.145	0.687 ± 0.143
	f1-score	0.682 ± 0.139	0.653 ± 0.152	0.711 ± 0.178	0.654 ± 0.179	0.707 ± 0.159	0.698 ± 0.143	0.686 ± 0.142
Sigma_PLV	accuracy	0.613 ± 0.124	0.675 ± 0.125	0.659 ± 0.125	0.591 ± 0.120	0.677 ± 0.117	0.629 ± 0.136	0.656 ± 0.131
	precision	0.619 ± 0.125	0.680 ± 0.126	0.662 ± 0.126	0.597 ± 0.122	0.681 ± 0.117	0.632 ± 0.137	0.659 ± 0.133
	recall	0.616 ± 0.122	0.675 ± 0.121	0.657 ± 0.122	0.595 ± 0.121	0.675 ± 0.114	0.631 ± 0.136	0.656 ± 0.130
	f1-score	0.611 ± 0.122	0.670 ± 0.124	0.653 ± 0.124	0.589 ± 0.121	0.672 ± 0.118	0.626 ± 0.136	0.655 ± 0.130
L_beta_PLV	accuracy	0.663 ± 0.148	0.628 ± 0.134	0.679 ± 0.163	0.595 ± 0.132	0.655 ± 0.127	0.642 ± 0.137	0.641 ± 0.183
	precision	0.663 ± 0.144	0.635 ± 0.145	0.686 ± 0.162	0.604 ± 0.137	0.659 ± 0.126	0.641 ± 0.137	0.642 ± 0.186
	recall	0.663 ± 0.144	0.631 ± 0.139	0.683 ± 0.161	0.603 ± 0.132	0.655 ± 0.123	0.643 ± 0.136	0.640 ± 0.186
	f1-score	0.659 ± 0.149	0.624 ± 0.136	0.675 ± 0.165	0.592 ± 0.135	0.653 ± 0.126	0.636 ± 0.140	0.636 ± 0.183
H_beta_PLV	accuracy	0.714 ± 0.143	0.680 ± 0.133	0.734 ± 0.145	0.675 ± 0.134	0.696 ± 0.141	0.719 ± 0.126	0.698 ± 0.137
	precision	0.715 ± 0.146	0.699 ± 0.136	0.738 ± 0.150	0.684 ± 0.137	0.701 ± 0.141	0.723 ± 0.129	0.708 ± 0.140
	recall	0.715 ± 0.147	0.688 ± 0.131	0.734 ± 0.149	0.680 ± 0.133	0.696 ± 0.143	0.722 ± 0.127	0.702 ± 0.136
	f1-score	0.710 ± 0.145	0.677 ± 0.132	0.732 ± 0.147	0.672 ± 0.131	0.691 ± 0.140	0.716 ± 0.126	0.696 ± 0.136

## Data Availability

Raw EEG data can be obtained by writing formal email to Fares Al-Shargie.

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
