# Peer review of "EEG Mental Stress Assessment Using Hybrid Multi-Domain Feature Sets of Functional Connectivity Network and Time-Frequency Features"

_sensors, 2021, doi:10.3390/s21186300_

Round 1
Reviewer 1 Report
The idea behind this paper is very interesting, i.e. try to explore the brain response evoked by mental stress with feature sets of Functional Connectivity Network and time frequency features. However, the paper needs considerable improvement before being publishable.
Some suggestions:
- Contribution. It seems that prior literature has already used EEG with machine learning method for similar applications such as driving fatigue or mental workload identification. Therefore, the authors need to clearly their contribution. Are there any new features developed for the mental stress prediction? or is the machine learning method novel and new? or is the prediction task substantially different from previous work? The paper contribution will be diminished if it only applies existing approach on a similar task.
- Moreover, I suggest the authors to show incremental effects of different feature combinations. More evaluation needs to be done along this line. I also suggest the authors to provide more details about the dataset and report precision, recall, and AUROC score.
- More information about the participants is needed in the study, i.e., age, gender, background and so on. The total number of the participants was 22. It seemed the sample size was too small and more participants need to be recruited. Appropriate sample size is important, and the number can be determined by some methods, e.g., G*Power. Moreover, gender affects many physiological measurements, the investigator must report how many of the subjects were male and female and must ensure that any group effects are not confounded by differences in the female/male ratios across groups. When studying normal subjects, the investigator should generally use either a similar number of female and male subjects or subjects of one gender only. The proportional imbalance in the female/male ratios may affect your results.
- Method section: it may be better to concise this section, especially provide reasons for your feature extracted but not just explain this features to readers.
- Line 235, page 6: “six features per EEG channel from the time domain”, while in line 249, page 8 “seven statistical features were extracted from”, so what are the actual number of features extracted from time domain?
- The chosen of sites to analyze was based on the visual inspection of the topographic maps. It seems arbitrary and needs the evidence from previous studies. Why did you choose these sites?
- Is there any physical load unexpectedly differed per condition, how do you deal with its effect? Did the experimental equipment affect participants’ brain response, how do you deal with this?
- It seems that Mental arithmetical tasks are most commonly used as stimuli for mental workload, is there any difference between stress and workload, how do you define this in your paper?
Reviewer 2 Report
This work assessed the 5 levels of stress on 22 healthy subjects using frontal Electroencephalogram (EEG) signals, Salivary Alpha-Amylase Level (AAL), and multiple machine learning (ML) classifiers. Multi-domain features were used for classification, including functional connectivity network estimated by Phase Locking Value (PLV) and temporal and spectral domain features. Overall, The novelty of this work is low. It looks like that the motivation is just no research add network features. It is too rough. We can not just add more and more types of features, because there are a mass of combinations and it is also hard to interpret the results. Meanwhile, the methodological contribution is still low. It is hard to follow the novelty of the proposed method. I would not like to suggest to publish this work on the journal.
[1] In the STRESS EEG MEASUREMENT AND PROTOCOLS, line 154, is there an incorrect use of dash?
[2] In STRESS EEG MEASUREMENT AND PROTOCOL, the author mentioned ‘For every rest/stress block, an arithmetic task pop-up for 30 seconds, followed by 20 seconds of rest.’, is this ‘rest’ different from the ‘rest condition’?
[3] There is a problem, the length of an epoch in this research is 1000ms that means 1 second (only 256 points), that is too short to extract the low frequency components like Delta band (frequency range is 1Hz-4Hz generally) effectively.
[4] There is only 7 channels, and average reference is not appropriate. Considering the potential effects of reference choices on the power indices and EEG networks, it is hard to explain or assess the results. That is, except obtaining the accuracy, there is few information to help us understanding the underlying brain mechanism.
[5] In the Time Domain Features (TDFs), the authors explained the time domain features by text and formulas. There should be a formula below the feature ‘Line length’. And the author should explain the meaning of letter σ, since the letter σ appeared in the previous chapter to describe a frequency band. Equation (10) is wrong, it should be k(k-1)/2.
[6] Although this research achieved high accuracy with fewer channels by fusing multi-domain features, there is still the problem of too large feature space?
[7] The resting state data may be calculated to determine whether these features only work in stress level classification to increase confidence of the results.
Reviewer 3 Report
- While there were innovative aspects to the paper, I feel that there was so much that the authors are trying to achieve that it wasn’t easy to be sure what the outcomes for this paper should be. The goals (aims) were never explicitly stated, and the authors need to think about this clearly so that the rest of the paper can flow better. What is it that you want the readers to get from this paper? There were contributions stated but not an aim. Please insert this and make it clear and then re-do the paper to have this focus in mind. Provide justifications for your study not just for the sake of novelty eg why was functional connectivity a feature? And many more.
- Line 66 to 134 of introduction: please summarise this to be concise (delete the majority of it)- what is the main problem? Right now, its long and unnecessary. You need 2 paragraphs, one that states the problem, and a second paragraph that leads to the aim of the study (how the findings will add to the literature and resolve the problems).
- Section 2.2 the protocol, was supposed to be a major contribution from this paper. Please rewrite this whole section to be more methodology like. It is currently one whole block and very tedious and long and repetitive. Be methodical, so that others can follow your protocol if they choose to replicate your study. Just clearly state the steps for the protocol, there rest can be deleted to avoid confusion (eg lines 148 to 155 consider deleting). Right now some important information-like description for rest, time stress etc are all hidden, these need to stand out more.
- Section 2.3 I did not understand what the authors were saying. Probably because section 2.2 on AAL only consisted of one sentence. Given that this is pretty much the one thing that is setting you apart from the many EEG and stress papers I think some more detail here is appropriate. I think section 2.3 is supposed to give the readers an idea of how all your data gets blended together and used, but I do not know what you are describing. This seems important to me, please get this section right.
- Lines 205-213- consider deleting. Too repetitive.
- 2 state the frequencies used eg theta (3-7.5 Hz) as this can vary especially things like high beta- what did you define as high beta?
- I can’t understand the purpose of 7 classifiers? If its just for the sake of reporting then please just choose the one most appropriate for your purpose. For example, maybe the classifier that handles hybrid features best. Right now, there is way too much reported and much of it feels repetitive. It adds to too much information with no real purpose.
- The connectivity results are also supposed to be unique to this paper. This is lost in the mess, and I find myself forgetting that it was even done. Why so many other features? Which feature is most important, otherwise, how could this possibly be made into something that can be useful in real time?
- Given the above suggested changes, please also make changes to your discussion so that it answers your aim.
- What is MIST???? How was this suddenly in the discussion but not mentioned before?
- Please check for grammatical errors.
Minor changes (typographical/grammatical)
Abstract:
Line 2: delete “stress”
Line 5: change to stressors (plural) delete “a”
Introduction:
Line 23: sentence “the long-term effects of stress…” does not read correctly. Please re-write to blend into second sentence. Maybe “the long-term effects of stress not only impacts on health issues such as…but has economic consequences too. The economic losses can reach…”
Round 2
Reviewer 2 Report
The issues I concerned were still not addressed in the revised version of the manuscript, and I still would not like to publish this work on the journal.
Author Response
"Please see the attachment"

Reviewer 3 Report
I appreciate that the authors have included the aims and it reads much better now. Because the aims are more carefully thought out the rest of the paper makes more sense.
Author Response
Thank you again, for you valuable remarks and careful
feedback which helped to enhance this manuscript and its presentation.